# Assessment of the Sustainability Performance of Eco-Engineering Measures in the Mediterranean Region

Slobodan B. Mickovski [1,*], Alejandro Gonzalez-Ollauri [1], Craig Thomson [1], Caroline Gallagher [1] and Guillermo Tardio [2]

1   BEAM Centre, School of Computing, Engineering and Built Environment, Glasgow Caledonian University, Cowcaddens Road, Glasgow G4 0BA, UK; alejandro.ollauri@gcu.ac.uk (A.G.-O.); craig.thomson@gcu.ac.uk (C.T.); c.e.gallagher@gcu.ac.uk (C.G.)
2   School of Forestry Engineering and Natural Resources, Technical University of Madrid, 28031 Madrid, Spain; secretaria@aeip.org.es
*   Correspondence: slobodan.mickovski@gcu.ac.uk

**Abstract:** Eco-engineering has a crucial role in defining and achieving the sustainability credentials of a civil engineering project. Better eco-engineering practices would help better in reducing the adverse impacts on the environment and society, but also on the financial performance of the project. However, the assessment of the sustainability effects of eco-engineering strategies can be challenging, as the treatment of this topic has been neglected in the scientific literature. The challenges lie in balancing the project delivery objectives with the sustainable design that will ensure appropriate and satisfactory environmental and financial performance and deliver social benefits such as ecosystem services. In order to achieve better practice and advance the knowledge in the field, there is a need for broader analysis of completed eco-engineering projects applied at different spatio-temporal scales. The aim of this study was to critically analyse 23 eco-engineering case studies provided by the ECOMED project partners using a life cycle analysis through a single sustainability framework based on a relatively small set of key performance indicators (KPIs), which reflect the principles of sustainability, and which are not contextual for eco-engineering projects. The objectives of this study are twofold: (i) to highlight areas of best practice and potential enhancement in the application of eco-engineering strategies, and (ii) to propose refinement and enhancement of the existing framework with KPIs contextual to eco-engineering projects. The results of the study suggest that the feasibility, mobilisation, and the long-term stages of an eco-engineering project are the most sustainable project stages, while the award, construction, and monitoring stages could generally benefit from a range of enhancements including benefits stemming from double-loop learning and a common basis for the specification and quantification of the financial resources needed to apply eco-engineering strategies. The outcomes of this study will benefit decision makers and eco-engineering practitioners alike in terms of not only raising the sustainability profile of the projects they are involved in, but also in terms of more efficient and cost-effective application of eco-engineering strategies.

**Keywords:** sustainability; slope stability; eco-engineering; nature-based solutions; green infrastructure; resilience

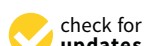



## 1. Introduction

Eco-engineering, defined as the proactive design of sustainable ecosystems which integrate human society with its natural environment for the benefit of both [1] is usually used broadly to describe long-term, ecological strategies to manage land with regard to natural or man-made hazards [2]. The integration of civil engineering techniques with natural or man-made materials to obtain fast, effective, and economic methods of protecting, restoring, and maintaining the environment [3] has long been recognised as a sustainable way of managing nature [4]. As in this approach, the use of vegetation for engineering

purposes gave rise to the concepts of 'green infrastructure' [5] and nature-based solutions (NbS; [6]).

The sustainability of the eco-engineering measures can be interpreted along the lines of a strong definition (based around the boundary limits between the sustainability dimensions) or a weak definition (based around accepting trade-offs between the sustainability dimensions; [7]). In either case, sustainability has an environmental, economic, and social dimension and the choice of definition is recognised to greatly influence decision making. While the three dimensions have usually been recognised when assessing the performance of any system, they may not be enough to assess the performance of eco-engineering measures which include vegetation that performs an engineering function which, usually, is providing engineering stability for the eco-engineering measure and the immediate environment [8]. Due to this, the assessment of the sustainability performance of the eco-engineering measures is even more challenging and has not been explored in detail in the past.

In the past, the various sustainability aspects of the eco-engineering measures have been assessed through cost evaluations, risk assessments, environmental impact assessments, and engineering calculations and modelling [8,9]. However, a standardised approach for the assessment of the sustainability performance of such measures in an integrated framework which captures the environmental, social, and economic dimensions of sustainability, on top of the engineering aspects, is lacking. [8] attempted to reconcile the traditional sustainability assessments with the assessment of engineering aspects of eco-engineering using a framework that considers the whole life cycle of an eco-engineering project (from inception to decommissioning). This framework envisages monitoring of a limited number of key performance indicators (KPIs) over the whole range of project stages as well as stakeholder cooperation not only throughout the project but also throughout the sustainability assessment.

Stakeholder cooperation and engagement, although well recognised as important (e.g., [10]) in eco-engineering decision-making, has not been considered in great detail in the assessment of sustainability performance. A number of trans-national and multi-disciplinary projects in Europe currently deal with this topic, which was of primary importance for the eco-engineering researchers and educators in the Mediterranean gathered around the ECOMED project (2017–2019; www.ecomedbio.eu; accessed on 4 April 2022). New alliances and dynamics between the stakeholders of an eco-engineering project were forged through interaction between the construction industry, professional bodies, academia, research institutions, and communities affected by the eco-engineering works [11]. Concepts such as the awareness of the effects of eco-engineering measures, education on all the aspects of sustainability of these measures, and the role and importance of double-loop learning [12] were explored through engagement of a multitude of stakeholders who participated in eco-engineering projects mainly based over the Mediterranean region.

The aim of this study was to critically analyse 23 eco-engineering case studies provided by the ECOMED project partners using a life cycle analysis through a single sustainability framework based on a relatively small set of key performance indicators (KPIs), which reflect the principles of sustainability and which are not contextual for eco-engineering projects. The objectives of this study were twofold: (i) to highlight areas of best practice and potential enhancement in the application of eco-engineering strategies, and (ii) to refine and enhance the existing framework with contextual KPIs through stakeholder consultation and engagement.

## 2. Methods

To achieve the objectives of this study, it was important to select representative eco-engineering projects across the Mediterranean which would demonstrate typical eco-engineering and, through analysis of the project life cycle, would reflect the rationale for undertaking these works by a range of stakeholders which would typically be involved

in such a project. The networks and partnerships forged through the ECOMED project were used to identify a broad range of stakeholders, firstly among the project partners, who were then consulted and asked to provide examples of eco-engineering studies from their respective countries. Bearing in mind that eco-engineering strategies have been relatively well described from the academic and scientific point of view (e.g., [2,13,14]), eco-engineering contractors and practitioners representing the industry in seven Mediterranean countries (within and outside the EU) were targeted to provide example case studies in order to ensure a broad cross-section of the markets and types of projects.

An initial questionnaire survey brief was prepared with an aim of highlighting the general groups/topics/scenarios of project information that will be needed for detailed analysis, as well as to check the availability and completeness of such information (Supplementary Material S1). The questionnaire survey was distributed to more than 500 contractors and eco-engineering practitioners across Europe through the European Federation for Bio-engineering. After the representative case studies were selected to ensure broad coverage of the countries across the Mediterranean where the ECOMED partners were located, more detailed surveys and semi-structured interviews with the relevant stakeholders were designed.

The detailed surveys were designed in the form of protocols [15] with an aim to outline more detail on each project stage and to include as many relevant sustainability themes/topics from each eco-engineering project. The detailed surveys for the selected case studies were carried out online using standardised forms and included mainly open-ended questions where the respondents could add relevant information but also include their perspective on any aspect of the project life cycle of eco-engineering measures.

In line with the ethical considerations in research, we anonymised each of the reviewed case studies including the names and affiliations of the surveyed and interviewed study participants. We considered each case study as part of a group rather than deriving results and conclusions from an individual source

After the detailed surveys were completed and the responses transferred to a secure electronic database, an initial review was carried out in order to establish the completeness of each survey as well as the perceived issues by the stakeholder and gauge their views on the future developments in eco-engineering. Where gaps in information were noted (e.g., little to no information on one or more project stages, design procedures, monitoring or testing arrangements, etc.) semi-structured in-depth interviews with the relevant stakeholders were carried out to extract rich data, including observational data from the case study information (Supplementary Material S2). For the purpose of this study, 15 in-depth semi-structured interviews were carried out with a range of stakeholders (contractors, practitioners, researchers, academics, engineers/site managers, community representatives, designers) for 16 out of the selected case studies.

Once the above were completed, an existing framework [12] was used to assess the sustainability performance of each of the selected case studies throughout their life cycle (all project stages from feasibility to long term impact). In this framework, the baseline sustainability score was 3.0 (mid-range value between 1.0—most detrimental to sustainability—and 5.0—most enhancement of sustainability, [12]), which was based on the average score of the KPIs for that particular stage, which meant that a project stage in a case study could be considered as sustainable only if the score for this project stage was above 3.0. This meant that each project stage for each of the case study projects was assessed for its sustainability performance and each case study was assessed for sustainability performance throughout its life cycle (eight project stages). For the purpose of this study, the project stages included (focusing on):

- Feasibility (including desk study; site investigation; bio-/geo-/hydro-assessment)
- Design (quantified risk assessment; construction management regulations; regulatory approvals; design options)
- Award (procurement process; pre-qualification; tendering; contractor credentials)
- Mobilisation (geographical location; amount of equipment; access; logistics; materials)

- Construction (materials; plant; energy; labour; water usage)
- Demobilisation (removal of equipment; mobilisation to next site)
- Monitoring (choice of instrumentation; monitoring categories; length of monitoring; invasiveness of monitoring process)
- Long term (effects on the immediate vicinity; regional effect(s); life cycle impact).

Basic descriptive statistics were then developed to reflect the above scoring to obtain the average score per project stage and the overall sustainability score for each case study. One and two-way ANOVA tests at 95% and 99% confidence level were undertaken to investigate statistical differences between the sustainability scores between project types, project stages, and country where the eco-engineering project is based. Firstly, suitability scores were statistically compared between project types. Then, statistical differences between project stages were evaluated in light of their corresponding sustainability score. For consistency, statistical differences between project stages were also evaluated for each project type individually. Eventually, score differences between project stages and country of origin were investigated. All the statistical tests were performed using the software R v3.5.1 [16] following normality testing through Shapiro–Wilk tests. Additional statistical analysis (Chi-squared/ANOVA) was performed to look for potential differences between different case study types (hill, river, coast environment). Both of the above were carried out to highlight the most and the least sustainable project stage(s) or case study/studies before justifying and supporting the result with the stakeholder opinions.

## 3. Results

From the responses to the initial questionnaire survey, 23 case studies were selected, the majority of which were located in the European Mediterranean (Figure 1) and represented three typical scenarios where eco-engineering is used for instability mitigation (mountain/hill slope, river, and coastal environments). Of the 23 selected case studies, 9 were in mountain/hill, 10 in fluvial, and 4 in coastal environment.

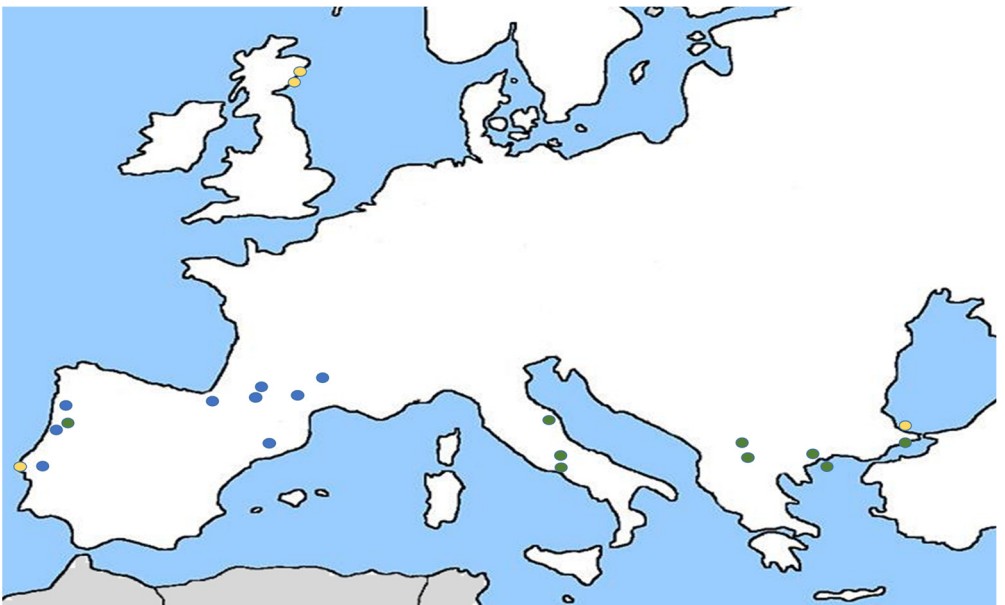

**Figure 1.** Geographical distribution of the case studies analysed in this study. Green circles: slope; blue circles: fluvial; yellow circles: coastal case studies.

The analysis of the detailed surveys and the scoring using the existing eco-engineering sustainability framework is shown in Figures 2–4, while the overall average sustainability performance for each project stage for all 23 case studies is shown in Figure 5.

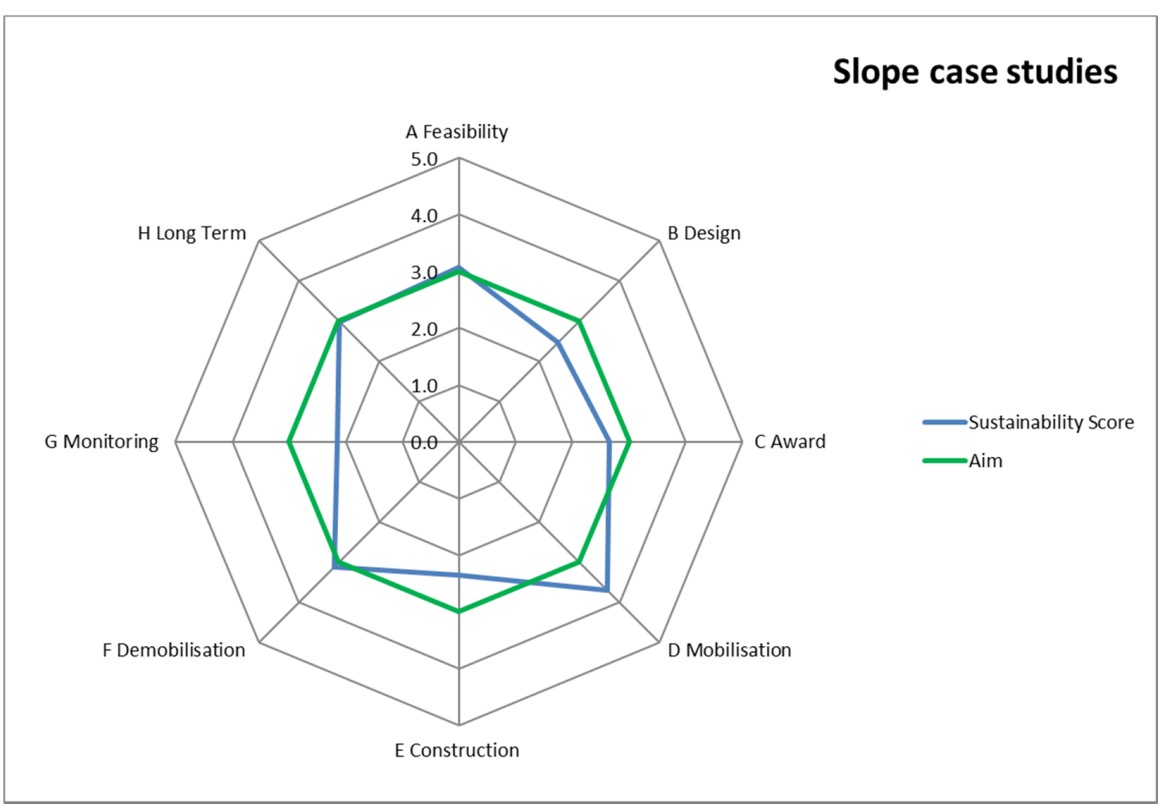

**Figure 2.** Average scores for hillslope case studies (9).

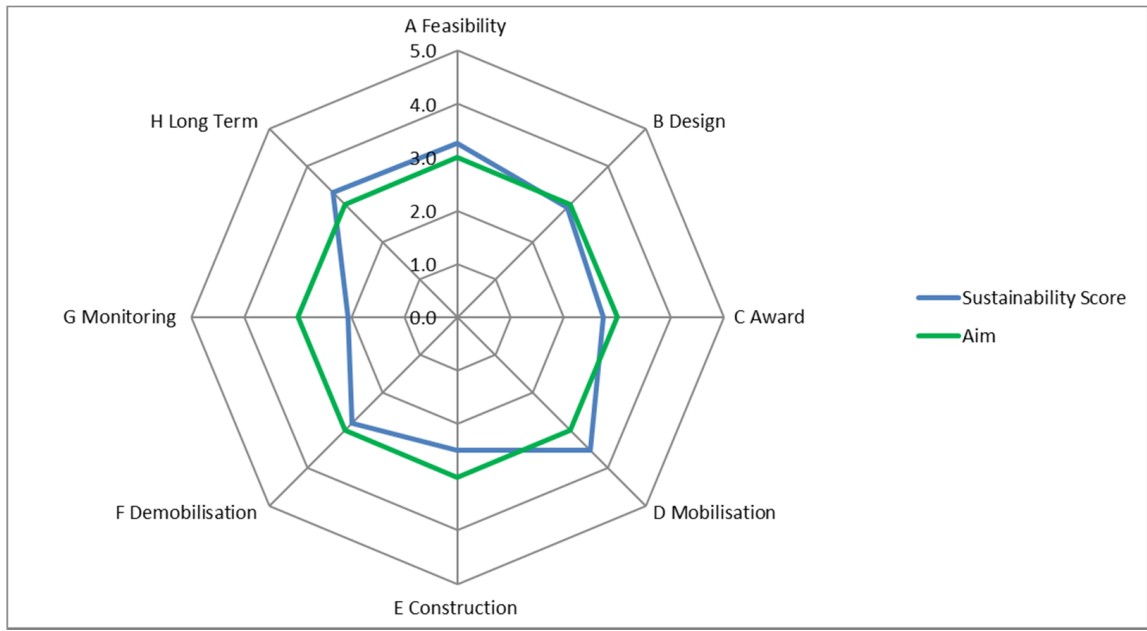

**Figure 3.** Average scores for fluvial case studies (10).

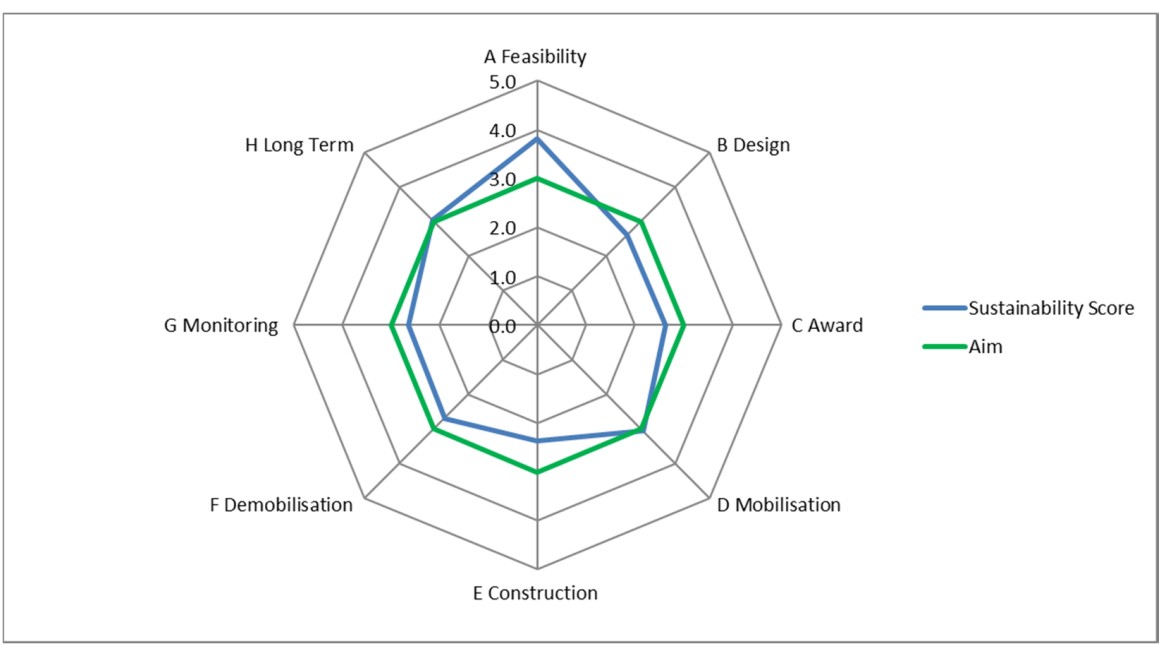

**Figure 4.** Average scores for coastal case studies (4).

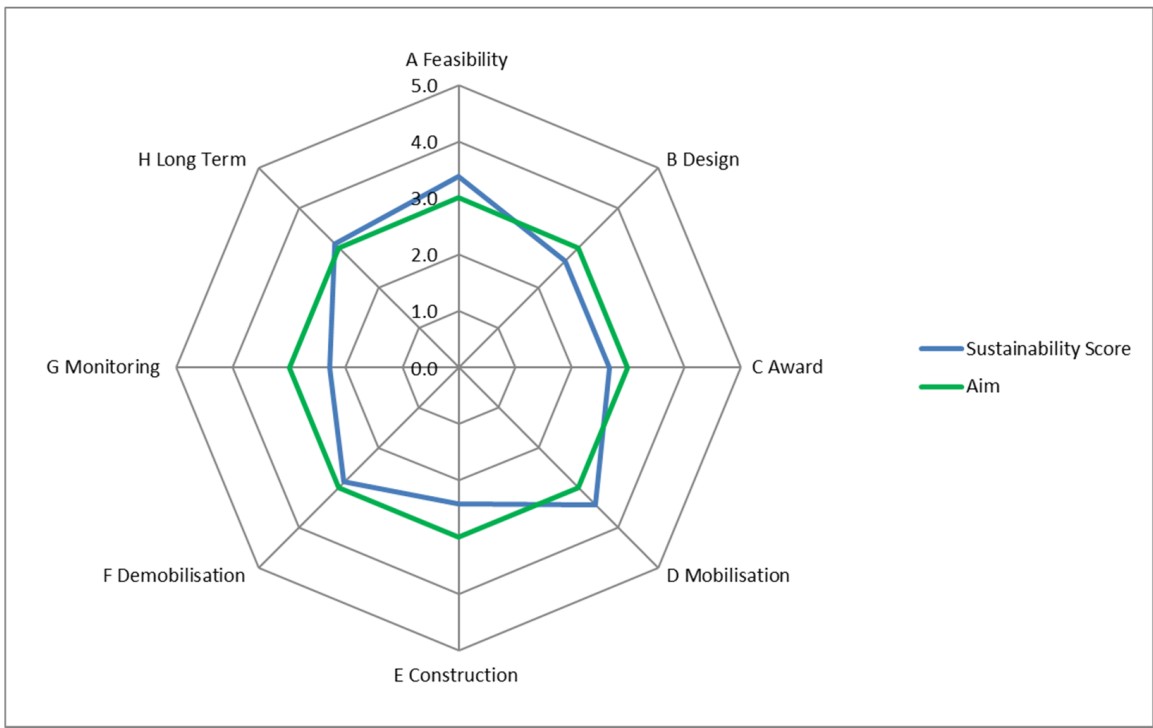

**Figure 5.** Average scores per project stage for all 23 case studies.

The average sustainability performance scores for hillslope case studies (9; Figure 2) showed that mobilisation to site was the most sustainable project phase, while the design, construction, and the monitoring phases were, on average, below the baseline level of sustainability for the project.

The average sustainability performance scores for the life cycle of the fluvial case studies (10; Figure 3.) showed that the feasibility, mobilisation, and long-term project phases were the most sustainable while, on average, the monitoring stage was the least sustainable.

The average sustainability performance scores for coastal case studies (4; Figure 4) showed that most of the project phases were at or just below the sustainability baseline performance with the construction phase score being on average the lowest for the project life cycle. This was opposed to the feasibility phase of the project which scored well above the sustainability baseline.

Average scores per project stage for all 23 case studies (Figure 5) showed that, overall, the selected eco-engineering case studies scored 2.85 ± 0.17 (cf. 3.00 as the sustainability baseline) and can be considered sustainable. This assessment was based on a very high average number of KPIs assessed per case study—108.77 ± 4.53 (out of 117 possible in the assessment framework)—which demonstrates the high level of data richness extracted from the survey and the interviews conducted for each case study.

When analysed individually within a project life cycle, the least sustainable project stages were shown to be the monitoring, construction, and award phases. The most sustainable project phases, on average, were feasibility, long term, and the mobilisation stages (Figure 6).

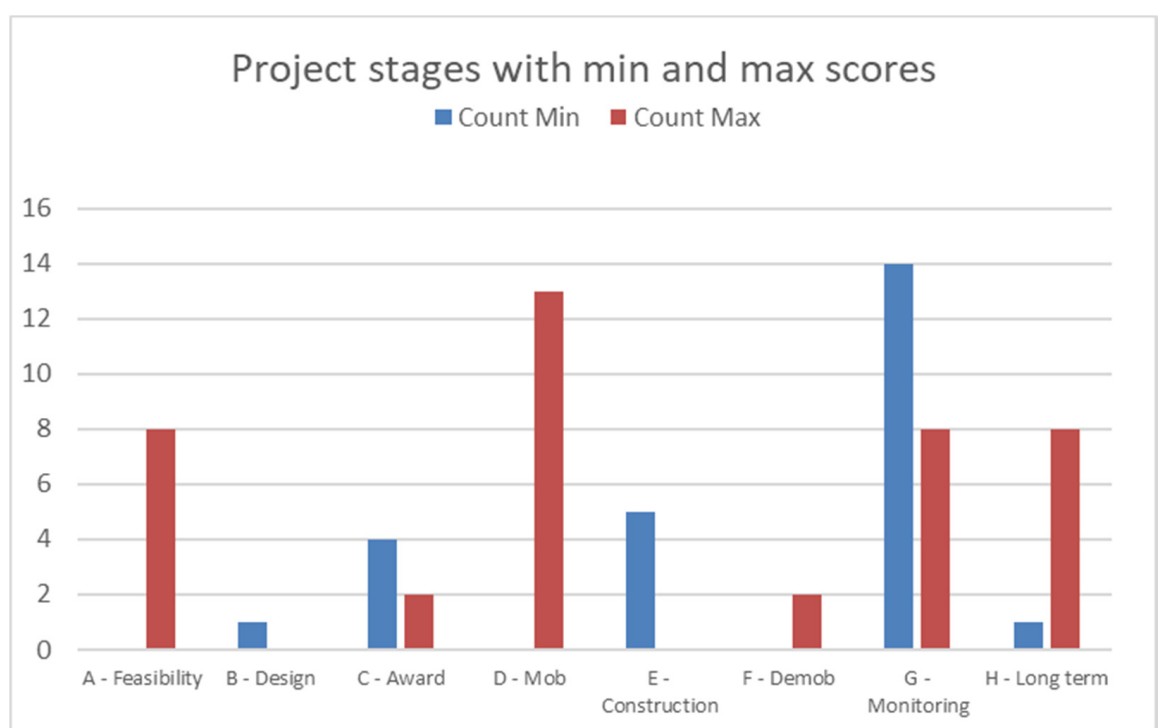

**Figure 6.** The least and most sustainable project stages as counted for each case study.

The overall average sustainability score per case study was not statistically different between coastal, fluvial, and slope case studies (F = 0.522, df = 2, *p* = 0.601; Figure 7), although, on average, the hillslope projects appeared to have a lower average sustainability score when compared to the two other types. However, the sustainability score was statistically different between project stages (F = 4.552, df = 7, *p* < 0.01; Figure 8). The most sustainable project phases, overall, were found to be mobilisation, feasibility, and long term considerations, while monitoring, construction, and award were found to be the least sustainable phases in the project life cycle.

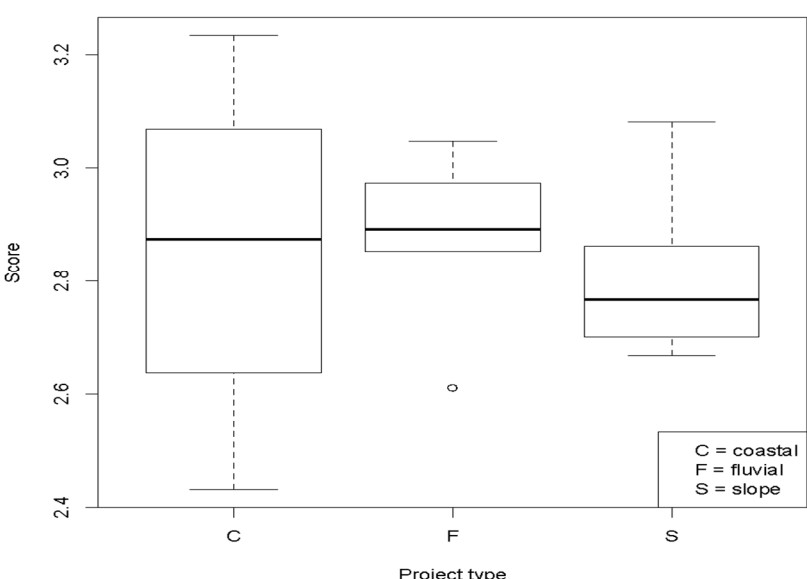

**Figure 7.** Overall average sustainability performance score per case study type. Baseline sustainability score is 3.0. The lower edge of the box corresponds to the 25th percentile data point, while the top edge of the box corresponds to the 75th percentile data point. The line within the box represents the median. The highest and lowest scores excluding outliers are shown in the upper and lower whiskers. An outlier was observed at ca. 2.6 in Fluvial projects (F).

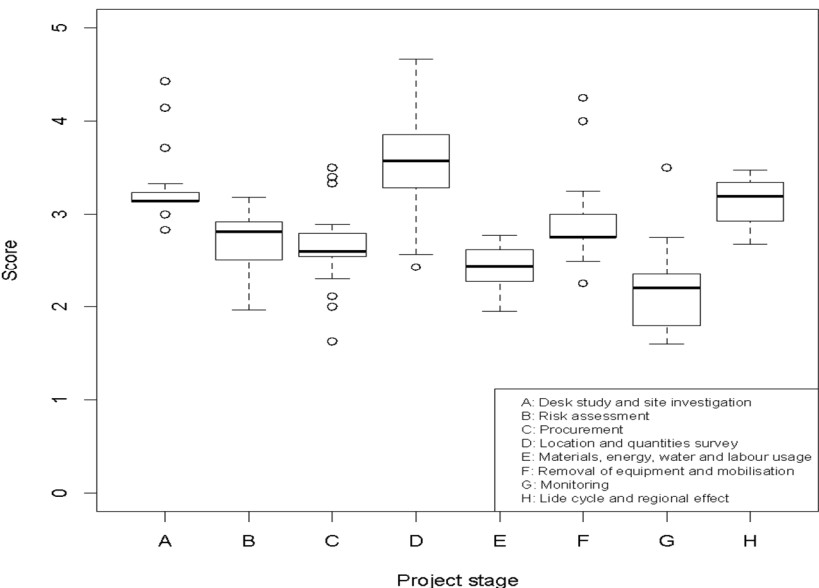

**Figure 8.** Overall average sustainability performance score per project stage. Baseline sustainability score is 3.0. The lower edge of the box corresponds to the 25th percentile data point, while the top edge of the box corresponds to the 75th percentile data point. The line within the box represents the median. The highest and lowest scores excluding outliers are shown in the upper and lower whiskers. The outliers in stages A, C, and F denote high score variability between the data points.

These differences were also consistent when they were evaluated for each project type individually (Coastal: F = 2.965, df = 7, $p < 0.05$; Fluvial: F = 37.57, df = 7, $p < 0.01$; Hill slope: F = 12.29, df = 7, $p < 0.01$; Figure 9.).

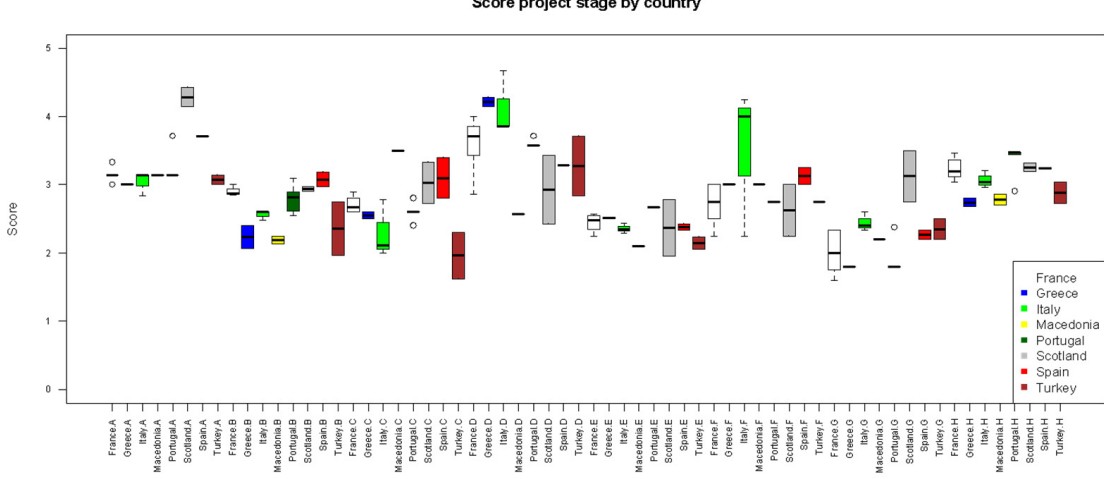

**Figure 9.** Differences between the average sustainability performance per project stage and type. C: coastal; F: fluvial; S: slope. The project stages are shown in Figure 8's legend. The lower edge of the box corresponds to the 25th percentile data point, while the top edge of the box corresponds to the 75th percentile data point. The line within the box represents the median. The highest and lowest scores excluding outliers are shown in the upper and lower whiskers. The outliers detected denote higher score variability in fluvial and slope projects than in coastal projects.

Still, the sustainability score of each project stage was not statistically different between project types (F = 0.698, df = 2, *p* = 0.499), but statistical differences were detected between the countries where the projects were based in (F = 2.775, df = 7, *p* < 0.01; Figure 10).

**Figure 10.** Differences between the average sustainability performance per project stage and country where the project is based in. The lower edge of the box corresponds to the 25th percentile data point, while the top edge of the box corresponds to the 75th percentile data point. The line within the box represents the median. The highest and lowest scores excluding outliers are shown in the upper and lower whiskers. Instances where only the median bar and outliers are portrayed denote a small dataset from which the statistical distribution could not be drawn.

## 4. Discussion and Analysis

The statistical analysis of the sustainability performance of 23 selected eco-engineering project case studies across the Mediterranean region showed that projects of this type can be considered sustainable when compared to a baseline sustainability as defined in the used assessment framework [17]. Very little difference between the case studies set in different environments was detected which shows that the eco-engineering project life

cycle is similar in different environments within several climatic zones. The latter suggests that eco-engineering projects follow a set of common non-standard protocols throughout their development but, more importantly, it highlights that eco-engineering projects are essentially sustainable which makes their application in practice viable across the range of climatic and geo-environmental conditions. The assessment of sustainability performance of eco-engineering measures should, however, be accompanied by an assessment of the sustainability performance of projects where traditional methods of construction (e.g., concrete or steel retaining walls, durable check dams, soil anchoring or nailing, etc.) have been used in order to highlight the benefits of using different materials and techniques in the various project stages and overall for the project. Such a comparison will contribute to the incorporation of the eco-engineering measures into the more standardised engineering approaches [14] and, in turn, the enhancement of the sustainability of the project while using the advantages of the different approaches.

The analysis of the sustainability performance of different project stages in different countries (Figure 10) showed that whether specific project stages perform above or below the sustainability baseline can and will depend on the local regulations and qualifications of the local labour force. Information gained from the semi-structured interviews that supports this result includes, for example, (a) the lack of monitoring measures and/or monitoring contract in almost all of the analysed case studies, (b) the non-existence of maintenance contract such as noted for the fluvial case studies in France where woodland management would have improved the sustainability efficiency of the installed measures, and (c) little environmental protection awareness and low level of environmentally friendly construction such as in a fluvial case study in Spain.

The analysis of project stage sustainability within the project life cycle showed significant differences between the project stages which were more sustainable and the project stages where sustainability performance can be enhanced by using best practice or standards. These differences could mostly be due to two general factors: (a) the state of the art in eco-engineering practice and (b) the set of KPIs in the framework used to assess the sustainability performance. The fact that the statistical analysis showed the least sustainable project stages to be award, monitoring, and construction phases was supported with the comments made by the stakeholders during the surveys and interviews. Namely, the lack of client awareness of the benefits of eco-engineering measures, together with the late inclusion of specialist eco-engineering contractors in the tendering process, were usually quoted as first obstacles for implementation of these strategies which affected the award stage sustainability performance in France and Spain. On the other hand, the fact that the eco-engineering companies are usually small, specialist contractors who often do not have full scale environmental management systems means that the quality submission accompanying the eco-engineering works would score less than the similar one for the implementation of traditional construction measures in a competitive commercial tendering process, which means the client would be likely to award the tender to a larger contractor who would construct traditional stabilisation measures (e.g., concrete and or steel-based works). Tendering processes should also include aspects related to public perception, environmental footprint, and the provision of co-benefits and ecosystem services to human communities, so eco-engineering companies can be more competitive at the tendering stage. The lack of uniformity in current approaches presents a real challenge when engaging with a construction sector which relies heavily on standardisation in its practices and to aid compliance through regulations and contracts. Standardisation in the assessment process through recognisable project stages has the benefit of fostering a shared understanding between stakeholders of the requirements for monitoring performance across the project, but instils the long-term perspective once the construction stages are complete. A set of standards and set of KPIs would provide the additional benefit of enabling direct comparison between projects and establishing a familiarity for stakeholders of the holistic nature of the framework. If applied too rigidly, this can provide a challenge in limiting projects to reflect their own contextual requirements and priorities through a tailored set of KPIs.

Despite this, [12] argue that given the low level of consideration of monitoring across the lifestyle of the intervention, then the absence of a common approach would benefit from a standardised approach and a framework of KPIs as long as project stakeholders can reflect their priorities through this approach. This can be used as a platform to facilitate understanding within the sector promoting a stronger consideration of sustainability through the life cycle and a requirement to achieve this through monitoring.

Similarly, the lower sustainability performance of the construction stage can be traced to the lack of consistent set of standards and specifications for eco-engineering works which feature among the existing set of KPIs and which, if developed in the future, would enhance and strengthen the eco-engineering practice. A viable alternative would be to develop new KPIs and incorporate the eco-engineering construction specifics [17] into the existing design and construction standards (e.g., Eurocodes; [14]). This alternative is preferred by several stakeholders who, during the surveys and interviews, reported lack of awareness of the environmental considerations within the traditional construction contractors who were implementing eco-engineering designs in Spain, Turkey, and France. Finally, almost all of the stakeholders who were surveyed and interviewed reported a lack of monitoring/maintenance works or contract accompanying the main eco-engineering designs. Although eco-engineering measures can be considered sustainable and often need little maintenance (e.g., [2]), monitoring in the short- and long-term is needed to verify the design and provide evidence of effectiveness of the implemented measures [18]. To enhance the sustainability performance of the project, monitoring of the KPIs relevant to the vegetation (e.g., [19]) can be specified in the form of monitoring protocols [20] which will accompany the design submission. In the same line, some KPIs from the existing framework, such as the length of the monitoring period and contingency planning, will need to be reformulated in order to encourage and help specifying monitoring when the framework is used as a planning tool.

The feasibility, mobilisation, and the long-term stages were identified as the most sustainable project phases, on average, for the analysed case studies. This shows that the eco-engineering designers have adopted the proactive approach when assigning an engineering function to the vegetation from the feasibility stage. The sustainability performance of this stage can be further increased by the development of eco-engineering standards for investigations/testing in situ and in the laboratory (e.g., [21]) which will make the design more specific and optimal in relation to the function assigned to the vegetation. Similarly, developing a quantified risk assessment [22,23] at this stage will enhance the design by incorporating the risks identified during the investigation/testing and will help with the design of the long-term measures. Additionally, some of the existing KPIs can be contextualised to take into account the specifics of the eco-engineering measures; for example, the KPI related to land which will be created for use by stakeholders will have to be reformulated in the cases where mitigation measures are implemented after a loss of land due to slope instability which will be restored but not specifically for use by general public or other stakeholders.

The high sustainability performance of the mobilisation stage is not surprising when the nature of the eco-engineering works is taken into account. The use of sustainable materials (usually seeds, plants, cuttings, biodegradable materials, etc.), combined with the very limited use of construction plant (only where structural works are to be undertaken) are the major reasons for the solid sustainability scoring. Further enhancements of the sustainability can be made in other project stages which affect mobilisation where, for example, planning and design specifications (and bills of quantities) can be used to minimise transport/deliveries to/from site, minimise waste, and thus enhance not only the sustainability of the mobilisation but also of the demobilisation stage of the project.

The sustainability performance of long-term considerations of the eco-engineering measures analysed in this study reflects the design for sustainability and resilience, bearing in mind that most of the analysed case studies were mitigation projects. The sustainability score of this project stage can be further enhanced with the inclusion of sustainability

benchmarking in the long term, but also verification of the long-term stability and the provision of (ecosystem) services in the long run. The long term stability of the eco-engineering measures can be verified by incorporating the specifics of the behaviour of the soil-vegetation-atmosphere continuum, including growth, load transfer, and synchronisation routines in the standard engineering design (e.g., [17,21,24]). The sustainability benchmarking in the long term and the provision of the long-term services can be achieved through enhancement of the applicability and relevance of the assessment framework. From this perspective, it is important to compile and analyse more projects that have gone through (almost) a full project cycle. Although the present study included five projects where corrective measures were applied after the original design was found insufficient and the benefits of double-loop learning ([12,25,26]) were recorded through the interviews, more projects need to be analysed with the existing framework in order to reflect the variety of eco-engineering works across Europe (and globally) and also to contextualise the problem-solving approach in eco-engineering where the applied measures were unsuccessful. This will contribute towards more focused KPIs which will reflect the opinions and experiences of various stakeholders [19] but also of the clients via completion of client satisfaction surveys which should be conducted in a structured manner [20]. Similarly, the existing framework can be enhanced with the incorporation of KPIs relevant to the ecosystem services [27–29]. These additional KPIs would be contextual with the eco-engineering measures and will be used for planning, as well as the assessment of the sustainability performance which will, in turn, strengthen the case for the eco-engineering project and enhance the sustainability of the award, construction, and monitoring stages.

## 5. Conclusions

Life cycle assessment of the sustainability performance of eco-engineering measures was performed for 23 case studies located, mainly, in the Mediterranean region. This assessment showed that the case studies, on average, performed near the baseline sustainability level (i.e., neither detrimental nor enhancing the sustainability). Future research should concentrate on the assessment of the sustainability performance of traditional land protection or restoration measures in order to critically compare sustainability performance between different approaches.

There were no statistical differences noted between the sustainability of eco-engineering projects in different environments (hill/mountain, river, coast) across the analysed case studies, although there were statistically significant differences in the sustainability performance of different project stages. In order to enhance the sustainability performance of different project stages and the eco-engineering project overall, the monitoring/maintenance efforts need to be enhanced starting from awareness in the feasibility stage, through contract and specification for the construction stage, and use of data in the long-term stages of the project. Future efforts should focus on the establishment of a consistent set of eco-engineering standards which would be applicable not locally but also regionally and trans-nationally. With this, an alignment of the different local regulations in terms of planning, contracting, and construction will be achieved while, at the same time, paving the way for compliance with the higher-level directives such as the EU Strategy on Green Infrastructure [30]. Additionally, these standards should include the baseline qualifications of the personnel involved in the planning, design, and construction of eco-engineering measures as well as the specification for the monitoring standards for eco-engineering measures, perhaps based on the existing handbook for practitioners [19].

The existing sustainability performance framework can be used confidently and fully for the assessment of eco-engineering projects with potential minor changes in several KPIs which may be needed in order to contextualise the specifics of different eco-engineering measures but also to include consideration of additional benefits (e.g., ecosystem services) or stakeholders (e.g., client satisfaction). The inclusion of sustainability criteria into the tendering process is a big factor in driving this whole agenda forward. Social value is also a clear criterion that could be considered in the context of eco-engineering, with a view to a

need to demonstrate the life cycle benefits. Standardisation in both the assessment process and the design will lead to a common understanding between stakeholders and also allow a platform for direct comparison between projects.

**Supplementary Materials:** The following supporting information can be downloaded at: https://www.mdpi.com/article/10.3390/land11040533/s1, S1: Questionnaire Survey template; S2: Semi-structured interview prompts template.

**Author Contributions:** Conceptualization, S.B.M.; methodology, C.T.; software, A.G.-O.; validation, C.G.; formal analysis, S.B.M. and A.G.-O.; investigation, G.T.; resources, S.B.M.; data curation, A.G.-O.; writing—original draft preparation, S.B.M.; writing—review and editing, C.T. and A.G.-O.; visualization, A.G.-O.; supervision, S.B.M.; project administration, C.G. and C.T.; All authors have read and agreed to the published version of the manuscript.

**Funding:** The study was funded by Erasmus+ project ECOMED (575796-EPP-1–2016-1-ES-EPPKA2-KA; www.ecomedbio.eu accessed on 4 April 2022).

**Institutional Review Board Statement:** Not applicable.

**Informed Consent Statement:** Informed consent was obtained from all subjects involved in the study.

**Data Availability Statement:** Publicly available datasets were analyzed in this study. This data can be found here: www.ecomedbio.eu accessed on 4 April 2022.

**Acknowledgments:** Thanks to J.L. Garcia, H. Celik, F. Gokbulak, G. Zaimes, Sangalli, Coronel y Assoc., Naturalea, Geing KuK, EcoSalix, I.C.E. Klaus Peklo, CBAG, JemmBuild, and AstroLabe.

**Conflicts of Interest:** The authors declare no conflict of interest.

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
