# Peer review of "Assessment of the Sustainability Performance of Eco-Engineering Measures in the Mediterranean Region"

_land, doi:10.3390/land11040533_

Round 1

Reviewer 1 Report

The manuscript:  Assessment of the sustainability performance of eco-engineering measures in the Mediterranean region approaches an important subject in Environmental Engineering. The paper fits the scope of the journal. The abstract presents a logical organization. The methodology is adequate.  The results are commented properly. Conclusions are justified by the results. The references given by the authors are sufficiently relevant. My recommendation is to accept the paper in present form.

Author Response

We thank the Reviewer for the positive comments and support. We have tried to enhance the  manuscript further by implementing the remarks requested by the other reviewers and we hope that this resulted in a better, more readable, manuscript.

Reviewer 2 Report

Interesting paper! If page limitation is not present, please add slight extension on the findings from the research. 
Other - mostly technical - comments can be found in the added file.

Author Response

We thank the reviewer for the positive and supportive comments. We have incorporated new text in the sections indicated by the Reviewer which resulted in, what we believe to be, a better manuscript which would convey the messages in a more comprehensible manner.

We thank the Reviewer for the comments and suggested edits. We have now incorporated all of the suggestions to the best of our knowledge.

Reviewer 3 Report

Following comments are furnished in order to improve the manuscript.

  1. ECOMED projects should be defined.
  2. Under Heading 2, title should be Methods
  3. Sample of  initial questionnaire survey template may be attached
  4. The template for the detailed surveys for the selected case studies  carried out online using standardised forms should be attached as the study is mainly based on the outcome of survey.
  5.  Sample Template for  semi-structured in-depth interviews with the relevant stakeholders may be illustrated.
  6. What was the basis of adopting  baseline sustainability score as 3.0 for sustainability?
  7. The details of 23 case studies has not been given. Only it has been mentioned that it broadly classified in different groups e.g.  9 
    were in mountain/hill, 10 in fluvial, and 4 in coastal environment.

Author Response

1. ECOMED projects should be defined.

Since this study is not focussing on the project itself, we have provided a reference to the ECOMED project (ln 80-86) which details the aim/objectives of the project and includes a web address where more details can be found.

2. Under Heading 2, title should be Methods

We have now incorporated this suggestion in the revised manuscript.

3. Sample of  initial questionnaire survey template may be attached

We have now included this in the supplementary material?

4. The template for the detailed surveys for the selected case studies carried out online using standardised forms should be attached as the study is mainly based on the outcome of survey.

The detailed survey protocols/templates have been published (Garcia Rodriguez, J. L., Sangalli, P., Tardio, G., Mickovski, S., Fernandes, J. P., & Gimenez, M. C. (Eds.) (2019). Specialization Process for the Bioengineering Sector in the Mediterranean Environment. Ecomed Project Part II. Protocols and Case Studies. 2019, 462 pp.) and can not be reproduced here due to the length and copyright. We, however, make a reference to these in the text.

5. Sample Template for semi-structured in-depth interviews with the relevant stakeholders may be illustrated.

Template for the questions/prompts/focus areas of the interviews is included in the supplementary material.

6. What was the basis of adopting  baseline sustainability score as 3.0 for sustainability?

The rationale has now been included in the text; it stems from assessing the sustainability effects across 5 categories ranging from 1 – most detrimental to sustainability, to 5 – most enhancing for sustainability, where 3 is neither detrimental nor enhancing (Mickovski and Thomson, 2017).

7. The details of 23 case studies has not been given. Only it has been mentioned that it broadly classified in different groups e.g.  9 were in mountain/hill, 10 in fluvial, and 4 in coastal environment.

In line with the ethical considerations in research, we have anonymised each of the reviewed case studies including the names and affiliations of the surveyed and interviewed study participants. We considered each case study as part of a group rather than deriving results and conclusions form an individual source. This clarification has now been inserted in the revised text of section 2. Methods.
